# Demonstration of Transmission Mode Soft X-ray NEXAFS Using Third- and Fifth-Order Harmonics of FEL Radiation at SACLA BL1

**Hiroshi Iwayama** [1,2,3,*] 🆔**, Masanari Nagasaka** [1,2,3] 🆔**, Ichiro Inoue** [3]**, Shigeki Owada** [3,4]**, Makina Yabashi** [3,4] 🆔 **and James R. Harries** [3,5]

1   Institute for Molecular Science, Myodaiji, Okazaki 444-8585, Japan; nagasaka@ims.ac.jp
2   SOKENDAI, The Graduate University for Advanced Studies, Myodaiji, Okazaki 444-8585, Japan
3   RIKEN, SPring-8 Center, Sayo, Hyogo 679-5148, Japan; inoue@spring8.or.jp (I.I.); osigeki@spring8.or.jp (S.O.); yabashi@spring8.or.jp (M.Y.); harries.james@qst.go.jp (J.R.H.)
4   JASRI, 1-1-1, Kouto, Sayo-cho, Sayo-gun, Hyogo 679-5198, Japan
5   QST, SPring-8, Kouto 1-1-1, Sayo, Hyogo 679-5148, Japan
*   Correspondence: iwayama@ims.ac.jp; Tel.: +81-564-55-7403

**Abstract:** We demonstrate the applicability of third- and fifth-order harmonics of free-electron laser (FEL) radiation for soft X-ray absorption spectroscopy in the transmission mode at SACLA BL1, which covers a photon energy range of 20 to 150 eV in the fundamental FEL radiation. By using the third- and fifth-order harmonics of the FEL radiation, we successfully recorded near-edge X-ray absorption fine structure (NEXAFS) spectra for Ar 2p core ionization and $CO_2$ C 1s and O 1s core ionizations. Our results show that the utilization of third- and fifth-order harmonics can significantly extend the available photon energies for NEXAFS spectroscopy using an FEL and opens the door to femtosecond pump-probe NEXAFS using a soft X-ray FEL.

**Keywords:** XFEL; soft X-ray absorption spectroscopy; high-order harmonics

## 1. Introduction

Free-electron lasers (FELs) [1] provide a source of intense, coherent, ultrafast, and continuously tunable radiation. In particular, since the development of the self-amplified spontaneous-emission (SASE) technique [2,3], which extended the available laser wavelengths to the X-ray regime [4–8], great interest has been shown by user communities over a wide range of fields in physics [9,10], chemistry [11,12], and structural biology [13,14].

X-ray FELs have made it possible to observe structural changes which occur at femtosecond timescales. For example, femtosecond bond formation was successfully observed by obtaining an X-ray diffraction image of three $Au(CN)_2^-$ molecules in solution using a 267 nm wavelength pump and X-ray FEL probe [15,16]. However, techniques using diffraction mainly require X-rays of energy 5 keV or higher, and heavy elements such as Au are required as scatterers in order to obtain sufficient signal strength. It is thus difficult to apply this method to biological molecules, which consist mainly of light elements such as carbon, nitrogen, and oxygen, and do not contain heavy elements.

The recent development of high-order harmonic generation (HHG) technology using lasers operating at wavelengths of 1 μm or longer has led to the availability of ultrashort pulsed light in the soft X-ray region [17]. The soft X-ray region of 200 to 1000 eV includes the K-shell absorption edges of light elements such as carbon (280 eV), nitrogen (400 eV), and oxygen (530 eV) and the L-shell absorption edges of transition metals such as chromium (580 eV), iron (720 eV), and cobalt (790 eV). Near edge X-ray absorption fine structure (NEXAFS) [18], a workhorse technique using

non-time-resolved synchrotron radiation, is sensitive to the local chemical environment of the specific atom targeted. By analyzing NEXAFS spectra, one can acquire information on the local structure, electronic, and spin state. Time-resolved NEXAFS in the soft X-ray region, especially above the carbon K-edge energy, is a promising tool and would enable the investigation of ultrafast structural, electron, and spin dynamics. Femtosecond time-resolved soft X-ray absorption spectroscopy by this HHG technology has been performed, and the photodissociation processes of $CF_4$ [19] and NO [20] molecules were observed. However, at present, usable light intensity can only be obtained for energies of up to about 400 eV (nitrogen absorption edge) [21]. Currently, the development of time-resolved soft X-ray absorption spectroscopy in the 500 to 1000 eV region is insufficient.

In 2020, at the XFEL facility LCLS in the United States, using soft X-ray absorption spectroscopy around the oxygen K-edge energy, it became clear that the generation of OH radicals by the chemical reaction of $H_2O^+$ ions ($H_2O^+ + H_2O \alpha \rightarrow OH + H_3O^+$) was about 46 femtoseconds [22], which attracted great interest from all over the world. Above 500 eV, the ultra-short pulses offered by an FEL are ideal for this application.

The SACLA BL1 FEL [23], targets the extreme ultraviolet (EUV) wavelength regime, providing photon energies ranging from 20 to 150 eV in the fundamental FEL radiation. Many successful applications have been made—for example, in EUV nonlinear optics [24,25] and ultrafast time-resolved spectroscopy [26].

In this paper, we propose and demonstrate the utilization of third- and fifth-order harmonics in FEL radiation to extend the available photon energy region. While the highest available photon energy is 150 eV in the fundamental FEL radiation at SACLA BL1, we demonstrate soft X-ray absorption spectroscopy at the Ar 2p and $CO_2$ C 1s and O 1s edges, successfully recording NEXAFS spectra by using the third- and fifth-order harmonics in the FEL radiation.

## 2. Methods

The experiments were carried out at SACLA BL1, which covers a photon energy range of 20 to 150 eV in the fundamental FEL radiation. The FEL radiation was linearly polarized in the horizontal direction, and the repetition rate was 60 Hz. A schematic diagram of the experimental setup for soft X-ray absorption spectroscopy in transmission mode with a gas cell is presented in Figure 1. The FEL pulses are transported and focused with a carbon-coated Kirkpatrick-Baez (KB) focusing mirror system [23]. The focus size was about 10 μm in diameter. The photon energy cutoff for the KB mirror system at an incident angle of 1.5° is roughly 1 keV, allowing soft X-ray absorption spectroscopy using the third- and fifth-order harmonics. In these experiments, we used photon energies of 82, 98, and 108 eV as the fundamental FEL radiation. In this energy region, pulse energies are roughly 100 μJ [23]. At a photon energy of 100 eV, the pulse energy of the third harmonic is estimated to be 0.3% [23] of the fundamental FEL radiation (i.e., around 300 nJ). According to the previous theoretical work [27], we estimate the pulse energy of the fifth-order harmonic to be around 0.01% of the fundamental FEL radiation (10 nJ). Full details of the beamline are described in [23]. Aluminum filters of thicknesses 0.5, 0.4, and 1.4 μm were used to selectively absorb the fundamental FEL radiation and optimize signal-to-background ratios of the Ar 2p, $CO_2$ C 1s, and $CO_2$ O 1s NEXAFS spectra, respectively. For example, the 0.5-μm-thick filter provides transmissions of $0.14 \times 10^{-5}$ at 86 eV (fundamental) and $0.4 \times 10^{-2}$ at 246 eV (third order).

The gas cell confines sample gas between two 100-nm-thick $Si_3N_4$ membrane windows, and for the experiments described here we used an absorption length of 10 mm. To avoid nonlinear optical effects, the gas cell was positioned 400 mm upstream of the FEL focus point. The FEL beam size was estimated to be about 1.4 mm in diameter at the gas cell position. The sample gas pressures for the Ar 2p, $CO_2$ C 1s, and $CO_2$ O 1s NEXAFS measurements were 720, 350, and 580 Pa, respectively.

Spectra of radiation transmitted through the gas cell were recorded using a flat-field imaging grazing incidence spectrometer (XUV639, manufactured by Shinkukogaku Co. Ltd., Tokyo, Japan), which was set on the same axis as the incident FEL axis. The FEL pulse was dispersed by an

aberration-corrected concave grating with a groove density of 600 lines/mm and focused on the detection plane of a thermoelectrically-cooled CCD camera (DO420-BN, Andor Technolgy Ltd). Assuming that 1 electron-hole pair can be made per 3.6 eV [28,29], the CCD intensity was converted to the number of photons detected. Error bars in absorption spectra were estimated from statistical errors of the number of photons detected.

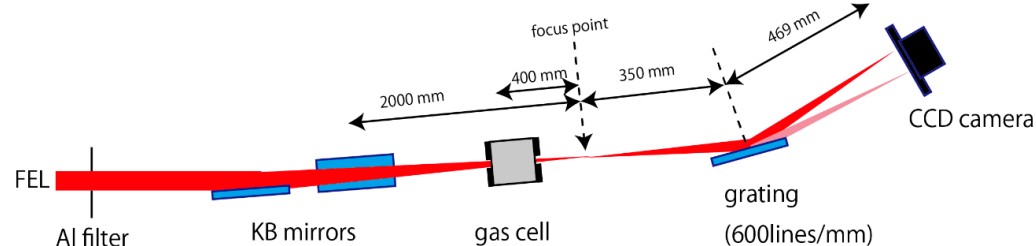

**Figure 1.** Schematic drawing of the experimental setup for soft X-ray absorption spectroscopy in transmission mode at SACLA BL1.

## 3. Results

NEXAFS spectra can be generated from our transmission mode spectra using the Lambert–Beer law, $\ln(I_0/I)$, where $I$ and $I_0$ are the transmission spectra recorded with and without sample gas, respectively. Figure 2a shows three typical FEL pulse spectra ($I_0$) recorded at BL1. The photon energy distributions have spike structures, and are different for each shot. These instable spectra may be attributed to the SASE-FEL which starts from shot noise in the electron beam. These spike structures are also observed for third- and fifth-order harmonics in the FEL radiation. In Figure 2b, the evolution of the spectrum as a function of averaging is shown. For 100 shots or less, the spike structure remains, but it can be seen that sufficient averaging is provided by averaging over 500 shots or more. Figure 2c shows 1000-shot average spectra. The bottom three spectra were measured at 3 min intervals, and the upper spectrum was recorded 66 min after the "0 min" spectrum. Since a thicker aluminum filter was used to record these spectra, it has been scaled to facilitate comparison. While the photon energy distributions are different for each shot, the average spectra are very similar and have a high reproducibility. There was little change in central light energy and width, even at 1 h apart. With the current experimental setup, we are unable to record both $I$ and $I_0$ shot-by-shot, and here we could only compare multiple-shot average $I$ and $I_0$ recorded consecutively. For soft X-ray absorption spectra of Ar and $CO_2$ gas shown below, we averaged 3000 shots for each spectrum.

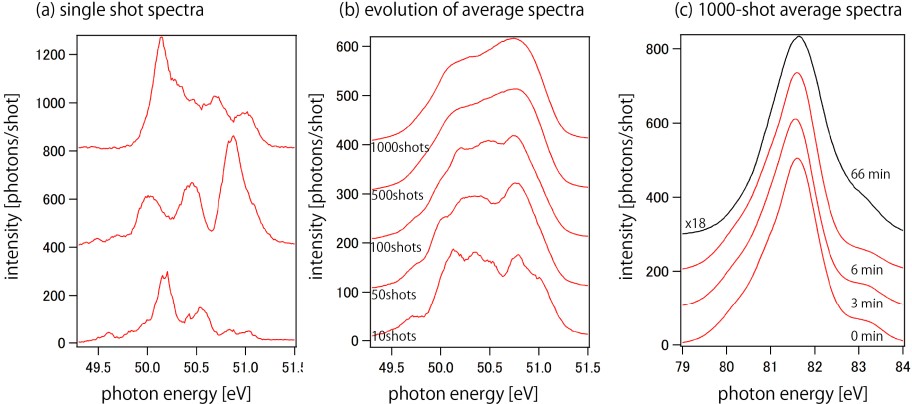

**Figure 2.** Typical (**a**) single-shot spectra, (**b**) the evolution of the spectrum as a function of the number of averaging, and (**c**) 1000-shot average spectra of incident free-electron laser (FEL) pulses.

Figure 3a shows average incident ($I_0$) and transmitted ($I$) spectra for Ar, recorded near the 2p edge. The photon energy of the FEL pulse was set to 82 eV, giving a third-order energy of 246 eV.

While the intensity of the third-order harmonics is only around 0.3% [23] of the fundamental FEL radiation, the transmission geometry allows us to directly record the transmitted FEL pulses. For the transmitted spectrum I, several sharp dips can be seen. These correspond to core excitations of Ar 2p → nl Rydberg states. Using the Lambert–Beer law, we can obtain the transmission mode absorption spectrum from $\ln(I_0/I)$, which is shown in Figure 3b. The sharp peaks correspond to Ar 2p → nl core excitations, and the overall structure of the absorption spectrum is very similar to that reported in the previous work [30]. The observed peak width of $2p_{3/2}$ → 4s at 244.4 eV is 200 meV, whereas the natural width is 116 meV [30]. We can thus estimate the resolution E/ΔE of our spectrometer to be around 1200. The spectral width of the FEL pulses is around 3% (ΔE/E), corresponding to 8 eV at a photon energy of 246 eV. This energy width allows us to record a NEXAFS spectrum without changing the FEL conditions or scanning the photon energy.

We also recorded the carbon and oxygen K-edge NEXAFS spectra of $CO_2$ molecules using the same technique. For the C 1s spectra, we set the FEL pulse energy to 98 eV, and used the third order (294 eV). For the O 1s spectra, we used 108 eV and the fifth order (540 eV). Figure 4 show the resulting C 1s and O 1s NEXAFS spectra. The strong peaks at photon energies of 291 and 533 eV are due to carbon and oxygen 1s → π* core excitations. The spectra also show C 1s → 3s Rydberg and O 1s → σ* core excitations. Due to the lower cross-sections and the lower photon intensity, the spectra are of lower quality than that recorded for Ar 2p. Both the $CO_2$ C 1s and the $CO_2$ O 1s spectra are similar to previous work [18,31]. The signal-to-noise ratio for $CO_2$ C 1s and the $CO_2$ O 1s spectra could be improved to match the Ar 2p spectrum by increasing the number of spectra used on average by a factor of 10. The 7th order of FEL pulses centered at 76 eV were too weak to be used to record O K-edge NEXAFS spectra.

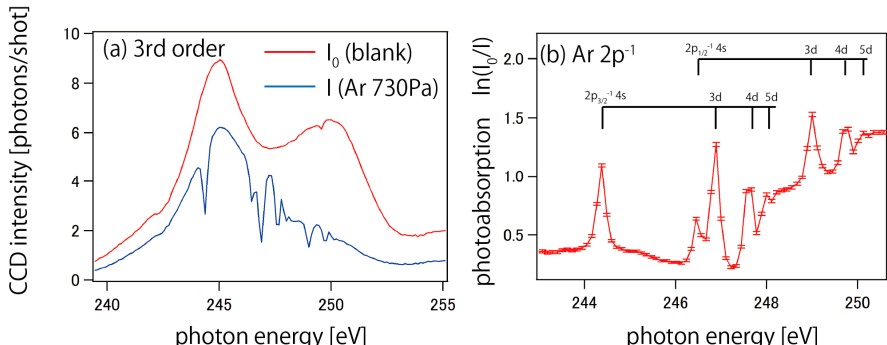

**Figure 3.** (**a**) Incident ($I_0$) and transmitted (I) 3000-shot average spectra for Ar gas. The photon energy of the incident FEL pulses was 82 eV in the fundamental FEL radiation. (**b**) Absorption spectrum for Ar gas obtained from $\ln(I_0/I)$.

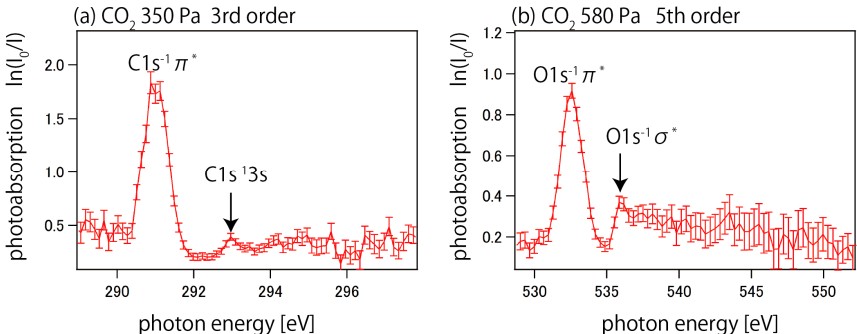

**Figure 4.** Soft X-ray absorption spectra for $CO_2$ gas around the (**a**) C 1s and (**b**) O 1s core ionization Table 98 and 108 eV, respectively.

## 4. Summary

We proposed and demonstrated the utilization of the third- and fifth-order harmonics in FEL radiation for soft X-ray absorption spectroscopy in transmission mode. We successfully recorded Ar 2p, $CO_2$ C 1s, and $CO_2$ O 1s NEXAFS spectra using the third- and fifth-order harmonics of the FEL radiation. The use of fifth-order harmonics extends the upper range of available photon energies at SACLA BL1 from 150 to 750 eV, covering the K-edge energies of light elements such as carbon, oxygen, and nitrogen, and the L-edge energies of transition metals. Since, especially at 500eV and above, HHG technology does not currently provide sufficient light intensity, soft X-ray FELs are the only available light source for soft X-ray absorption spectroscopy around the oxygen K-edge energy. By combining with a synchronized visible or infrared laser, pump-probe experiments will enable time-resolved soft X-ray absorption spectroscopy at SACLA BL1.

**Author Contributions:** H.I., M.N. and J.R.H. designed the experimental platform and performed experiments. H.I. analyzed the data. H.I. and J.R.H. wrote the paper. I.I., S.O. and M.Y. calibrated and controlled the beamline of SACLA BL1. All authors have read and agreed to the published version of the manuscript.

**Funding:** This research received no external funding.

**Acknowledgments:** The experiments were performed at SACLA BL1 with support of SACLA Basic Development Program. The authors would like to acknowledge the supporting members of the SACLA facility.

**Conflicts of Interest:** The authors declare no conflict of interest.

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
