# Peer review of "Demonstration of Transmission Mode Soft X-ray NEXAFS Using Third- and Fifth-Order Harmonics of FEL Radiation at SACLA BL1"

_applsci, doi:10.3390/app10217852_

Round 1

Reviewer 1 Report

The introduction presented is too short and general.There is no comprehensive introduction to the topic. In  my opinion, the introduction should be significantly expanded. The drawing and its description should be placed another section called methods and materials. The summary is quite short and general. In my opinion, the article is too short and should be significantly expanded and edit.  I suggests increasing the amount of research and reporting more cases. The aticle may  be published after the content has been extended.

Author Response

Response to Reviewer 1 Comments

Dear Referee 1,

We heartily thank the referees for his/her valuable comments on our work and for his/her very useful suggestions. We have taken them all into account, and we feel that the paper is now significantly improved. Added and modified sentences and words are highlighted in yellow. Main points of the revision are as follows.

  • In introduction, we added research examples of ultrafast spectroscopy with XFEL and HHG and explained reasons why we need soft x-ray FEL.
  • From a statistical error of the number of photon detected, we estimated errors and added error bars in absorption spectra.

Point 1: The introduction presented is too short and general. There is no comprehensive introduction to the topic. In my opinion, the introduction should be significantly expanded. The drawing and its description should be placed another section called methods and materials. The summary is quite short and general. In my opinion, the article is too short and should be significantly expanded and edit.  I suggests increasing the amount of research and reporting more cases. The aticle may  be published after the content has been extended.

In the introduction section, we added research examples of ultrafast spectroscopy with XFEL and HHG and explained the reasons why we need soft x-ray FEL. We also modified all figures and increased information on the experiments. The amount of the article was significantly increased.  

Reviewer 2 Report

Dear Authors and Editor,

in the manuscript "Demonstration of transmission mode soft X-ray NEXAFS using higher-order harmonics in FEL radiation at SACLA BL1" by Iwayama et al., the Authors have nicely described their development of harmonic generation using the free electron laser (FEL). Their technique is of significant interest in view of the general application of FELs and the future time-resolved experiments. The manuscript could be published if the following issues can be addressed satisfactorily:

(1) In this manuscript only low-order harmonics up to the 5th order have been demonstrated. Therefore it could be hard to justify the terminology "higher-order harmonics" here. The Authors need to consider the correct usage of this terminology and modify the title as well as the content of this manuscript.

(2) In figure 2 both the x- and y-axis need to be labelled with the correct scales and units. More importantly, the evolution of the spectrum as a function of the number of averaging should be shown.

(3) In figures 3 and 4, the data points need to be presented clearly, including the indication of error bars for the intensity as well as the energy.

(4) In the attached PDF file some further minor corrections can be found.

Author Response

Response to Reviewer 2 Comments

Dear Referee 2,

We heartily thank the referees for his/her valuable comments on our work and for his/her very useful suggestions. We have taken them all into account, and we feel that the paper is now significantly improved. Added and modified sentences and words are highlighted in yellow. Main points of the revision are as follows.

  • In introduction, we added research examples of ultrafast spectroscopy with XFEL and HHG and explained reasons why we need soft x-ray FEL.
  • From a statistical error of the number of photon detected, we estimated errors and added error bars in absorption spectra.

Point 1: In this manuscript only low-order harmonics up to the 5th order have been demonstrated. Therefore it could be hard to justify the terminology "higher-order harmonics" here. The Authors need to consider the correct usage of this terminology and modify the title as well as the content of this manuscript.

We replaced all terminology of “higher-order harmonics” with that of “third- and fifth-order”.

Point 2: In figure 2 both the x- and y-axis need to be labelled with the correct scales and units. More importantly, the evolution of the spectrum as a function of the number of averaging should be shown.

In Fig. 2, we labelled x- and y-axis with the correct scales [photons/shot]. And we also added a new figure which shows the evolution of the spectrum as a function of the number of averaging.

Point 3: In figures 3 and 4, the data points need to be presented clearly, including the indication of error bars for the intensity as well as the energy.

We estimated errors in absorption spectra from statistical errors of the number of photons detected. We added error bars in Fig. 3 and 4.

Point 4: In the attached PDF file some further minor corrections can be found.

Thank you for putting so much effort into proofreading my sentences. I really appreciate it. In Fig. 1, we added descriptions of distances between optics.

Reviewer 3 Report

The paper by Iwayama et al. concisely and convincingly demonstrates that the 3rd and 5th harmonics of the SACLA BL1 FEL are useful for measuring NEXAFS spectra. Although the shot-to-shot spectrum of the FEL is very noisy, the averaged spectrum is very stable, and useful for absorption spectroscopy.

Even in the 5th harmonic, 10 nJ/pulse is available, which is more than enough for Xray absorption spectroscopy. The results will be of interest to anyone considering proposing an ultrafast Xray absorption experiment, and definitely merit publication.

I have a few specific questions:

  1. How good is the long term stability of the spectrum? In Fig. 2 we see three different 1000shot averaged spectra, all of which look very similar. How much time passed between recording these spectra, and over what timescale do you expect it to remain stable? This is particularly relevant when it comes to time-resolved measurements, where the system must remain stable over a long scan.
  2. Would it be possible to add an x-scale to Fig. 2? I am curious as to what the spacing of the modulations looks like on the frequency axis.
  3. The Ar spectra (Fig 3) look very nice, but the CO2 spectra (Fig 4) have much higher noise, which is attributed to lower absorption cross sections, and reduced photon flux. Were these spectra also recorded with 3000 shots? Do you have a feel for how many shots would be needed to bring the data up to the quality of the Ar spectrum?
  4. How does this technique compare to tabletop high harmonic generation for NEXAFS? HHG can also produce femtosecond pulses in the soft X-ray regime, with enough intensity and stability to measure absorption spectra. I am interested as to you opinions of the pros and cons of these two different ways of measuring femtosecond NEXAFS spectra.

Author Response

Response to Reviewer 3 Comments

Dear Referee 3,

We heartily thank the referees for his/her valuable comments on our work and for his/her very useful suggestions. We have taken them all into account, and we feel that the paper is now significantly improved. Added and modified sentences and words are highlighted in yellow. The main points of the revision are as follows.

  • In the introduction, we added research examples of ultrafast spectroscopy with XFEL and HHG and explained the reasons why we need soft x-ray FEL.
  • From a statistical error of the number of photons detected, we estimated errors and added error bars in absorption spectra.

Point 1: How good is the long term stability of the spectrum? In Fig. 2 we see three different 1000shot averaged spectra, all of which look very similar. How much time passed between recording these spectra, and over what timescale do you expect it to remain stable? This is particularly relevant when it comes to time-resolved measurements, where the system must remain stable over a long scan.

We modified Fig. 2(c) and added the sentence of “The bottom three spectra were measured at 3-minute intervals, and the upper spectrum was recorded 66 minutes after the ’0 min’ spectrum.” At SACLA BL 1, there was little change in central light energy and width, even at 1 hour apart.

Point 2: Would it be possible to add an x-scale to Fig. 2? I am curious as to what the spacing of the modulations looks like on the frequency axis.

We added x- and y-axis in Fig. 2. The CCD intensity was converted into the number of photons detected by assuming that 1 electron-hole pair can be made per 3.6 eV.

Point 3: The Ar spectra (Fig 3) look very nice, but the CO2 spectra (Fig 4) have much higher noise, which is attributed to lower absorption cross sections, and reduced photon flux. Were these spectra also recorded with 3000 shots? Do you have a feel for how many shots would be needed to bring the data up to the quality of the Ar spectrum?

To compare the signal-to-noise ratio of Ar and CO2 absorption spectra, we added error bars in Fig. 3 and 4. By comparing the numbers of photons detected, we found that we need 10 times more statistics for CO2 carbon and oxygen 1s absorption spectra.

Point 4: How does this technique compare to tabletop high harmonic generation for NEXAFS? HHG can also produce femtosecond pulses in the soft X-ray regime, with enough intensity and stability to measure absorption spectra. I am interested as to you opinions of the pros and cons of these two different ways of measuring femtosecond NEXAFS spectra.

At present, usable light intensity can only be obtained for energies of up to about 400 eV (nitrogen absorption edge) [21]. Currently, the development of time-resolved soft X-ray absorption spectroscopy in the 500 to 1000 eV region is insufficient. Soft x-ray FELs are the only available light source for soft x-ray absorption spectroscopy above the oxygen K-edge energy. So we believe soft x-ray FEL has an advantage in the soft x-ray region above 500 eV, where includes K-edge energies of oxygen and fluorine and L-edge energies of cobalt, iron, and chromium.

Round 2

Reviewer 1 Report

Thank you for considering your comments. In my opinion, the article is suitable for publication.

Author Response

We heartily thank you agina for your valuable comments on our work. By modifying the introduction section as you suggested, we feel that the paper is significantly improved.

Reviewer 2 Report

Dear Authors and Editor,

the manuscript has been significantly improved. However, the following issues still need to be addressed before publication:

(1) On page 3, near the end of the third paragraph, the Authors wrote: "Assuming that 1 electron-hole pair can be made per 3.6 eV, the CCD intensity was converted to the number of photons detected." The Authors need to explain directly, whether the value of 3.6 eV comes from the direct band gap of silicon. Moreover, the quantum efficiency of the CCD can depend strongly on the photon energy and should be considered explicitly.

(2) In the sentence directly following that in (1), the additional period at the end needs to be removed.

Author Response

Thank you very much for providing important comments again. We are thankful for the time and energy you expended.

point 1 On page 3, near the end of the third paragraph, the Authors wrote: "Assuming that 1 electron-hole pair can be made per 3.6 eV, the CCD intensity was converted to the number of photons detected." The Authors need to explain directly, whether the value of 3.6 eV comes from the direct bandgap of silicon. Moreover, the quantum efficiency of the CCD can depend strongly on the photon energy and should be considered explicitly

Answer:

We referred to the manual (datasheet) of the Andor Website. In the revised manuscript, we added two references (website and original paper). Above 100eV, the value of 3.6 eV is nearly constant.

While the quantum efficiency (detection efficiency) depends on the photon energy, error bars result come from the number of "detected" photons. Thus when we estimate error bars, the quantum efficiency does not matter.